# Probing the ultrafast dynamics of excitons in single semiconducting carbon nanotubes

Konrad Birkmeier[1,2], Tobias Hertel [3] & Achim Hartschuh [1] ✉

Excitonic states govern the optical spectra of low-dimensional semiconductor nanomaterials and their dynamics are key for a wide range of applications, such as in solar energy harvesting and lighting. Semiconducting single-walled carbon nanotubes emerged as particularly rich model systems for one-dimensional nanomaterials and as such have been investigated intensively in the past. The exciton decay dynamics in nanotubes has been studied mainly by transient absorption and time-resolved photoluminescence spectroscopy. Since different transitions are monitored with these two techniques, developing a comprehensive model to reconcile different data sets, however, turned out to be a challenge and remarkably, a uniform description seems to remain elusive. In this work, we investigate the exciton decay dynamics in single carbon nanotubes using transient interferometric scattering and time-resolved photoluminescence microscopy with few-exciton detection sensitivity and formulate a unified microscopic model by combining unimolecular exciton decay and ultrafast exciton-exciton annihilation on a time-scale down to 200 fs.

Owing to their quasi-one-dimensional structure, semiconducting single-walled carbon nanotubes (SWCNTs) exhibit especially strong electron–electron interactions and correlations, resulting in a range of different multi-particle phenomena. Excitons, i.e. tightly bound electron-hole states, are the origin of sharp optical resonances with energies that are directly determined by the nanotube's diameter and chirality[1-3]. Efficient exciton absorption and long exciton lifetimes are key for a variety of applications including solar energy harvesting, fluorescence labeling, and sensing[4-6]. The excited state dynamics of semiconducting single-walled carbon nanotubes has been widely explored predominantly using either time-resolved PL spectroscopy or transient absorption spectroscopy, two techniques that probe different transitions and potentially different electronic states, thereby providing contrasting perspectives.

Detecting spontaneous emission with time-correlated single photon counting (TCSPC), has been widely utilized to investigate the excited state dynamics of semiconducting SWCNTs on the ensemble level and of single nanotubes[7-10]. Typical lifetimes

of bright $E_{11}$ excitons in small-diameter SWCNTs are in the range of 10–70 ps. Besides providing information on the bright exciton lifetime, time-resolved PL spectroscopy revealed a complex interplay of processes involving the branching to long-lived dark excitons, exciton diffusion and end-quenching[7]. Single nanotube studies showed that the observed properties may vary from nanotube to nanotube for a variety of reasons. These include different concentrations of quenching sites caused by defects or local charge doping, quenching of mobile excitons at the nanotube end and the influence of a heterogeneous environment[7-9]. In TCSPC, however, the temporal resolution is limited by the response time of the detector and the electronics to about 10 ps in most implementations. Early ultrafast photoinitiated processes may thus remain hidden. Streak cameras provide a temporal resolution down to 1 ps in single photon counting mode. However, available systems have low detection efficiencies above 900 nm, the range of SWCNTs PL emission. Higher temporal resolution can also be achieved in femtosecond excitation correlation spectroscopy, the signal analysis

[1]Department of Chemistry and CeNS, LMU Munich, Butenandtstr. 5-13, 81377 Munich, Germany. [2]TOPTICA Photonics AG, Lochhamer Schlag 19, 82166 Gräfelfing, Germany. [3]Institute of Physical and Theoretical Chemistry, Julius-Maximilian University Würzburg, 97074 Würzburg, Germany. ✉e-mail: achim.hartschuh@lmu.de

and interpretation appear to be less direct requiring additional modeling of the signal formation process[11,12].

In contrast, transient absorption (TA) spectroscopy, for which the temporal resolution is only limited by the laser pulse duration, readily reaches into the 10 fs regime. Moreover, TA can directly probe non-emissive states, coherent phenomena as well as energy transfer processes[13–15]. Since the discovery of SWCNTs a large number of studies addressed the excited state dynamics of SWCNTs on the ensemble level. By probing transient changes in the sample's transmission and not relying on spontaneous emission, these studies revealed the dynamics of a larger series of photoinduced states including biexcitons and trions as well as triplet excitons[2,13,16–22]. In TA, the main signal contributions result from ground state bleaching related to phase-space filling, stimulated emission and absorption from photo-generated states. The occurrence of these different, spectrally overlapping contributions can render their clear distinction difficult, in particular for samples containing multiple nanotube species (n,m).

From transient absorption experiments distinctly different models for the decay dynamics of $E_{11}$ excitons have been reported and to date, a uniform description seems to remain elusive. Early transient absorption experiments on (6,5)-enriched samples seemed to suggest an initially mono-exponential decay of the $E_{11}$ excitons with a lifetime of 6 ps[16]. In ref. 18 the signal transients of undoped (8,3) SWCNTs in solution where described by two fast coupling rates to different dark states and a slower decay rate of 1/6.4 ps. Early transient absorption studies already pointed out the importance of multi-particle processes in terms of excitation intensity-dependent exciton-exciton annihilation (EEA)[23–29]. For very high fluences a cross-over between diffusion and reaction-limited EEA at about 3 ps was reported[25]. For longer time scales several TA studies describe the exciton dynamics only by $t^{-1/2}$ decay that results from the diffusive character of non-radiative decay at charged impurities[30] or two-particle interactions leading to EEA[19,25] without the exponential decay components as observed in time-resolved PL experiments[7–10].

Indeed, several studies find different behavior for transient absorption and fluorescence experiments which suggests that the transient species contributing to the pump-probe signal is different to that dominating the time-resolved fluorescence[23,28]. For example, whereas the signal transients obtained by transient absorption and time-resolved PL have been both explained by three-level systems including optically dark states, very different branching and relaxation rates have been derived[7,8,18] possibly reflecting the different timescales probed by these experiments. Other reports using transient absorption spectroscopy found tri-exponential decay dynamics on the sub ps to few tens of ps timescale from ensemble experiments in solution, which were attributed to phonon-assisted processes[31] with no direct counterpart in PL experiments.

By probing relative changes in signal intensities instead of a background-free signal as in the case of PL detection, transient absorption spectroscopy generally achieves lower detection sensitivities. Notably, ultrafast pump-probe spectroscopy has been demonstrated to be applicable to single molecules and semiconductor nanocrystals in special implementations[32,33] and has also been applied to SWCNTs with larger diameters in the range 1.1–2.1 nm[34]. In these experiments, the transient absorption signals were detected in the range of higher excitonic transitions ($E_{33}/E_{22}$) that are known to be short-lived as compared to $E_{11}$ excitons. The observed decay time of about 1 ps for all the semiconducting nanotubes studied was attributed to coupling between the excitons created by the pump laser pulse and the substrate[34].

From the discussion above, significant discrepancies between the models derived from time-resolved PL and transient absorption studies persist and a unifying picture simultaneously describing data from both spectroscopic probes by a single comprehensive model based on

the same transition rates appears to be lacking. This could be attributed to a number of reasons including the fact that PL and TA spectroscopy probe different transitions. For example, the presence of multiple (n,m) species can complicate the analysis, in particular in case of TA data with overlapping signal contributions. On the other hand, the lower time resolution of TCSPC might not be sufficient to reveal the very early excited state dynamics that dominate the TA response. Finally, different excitation fluences might have been applied in the experiments—TA measurements are often carried out at higher fluences—which will lead to varying contributions of EEA.

Here, we investigated single (6,5) SWCNTs, possibly the most studied species, using confocal transient interferometric scattering (TiSCAT) and time-resolved PL microscopy. TiSCAT enables the sensitive detection of the population dynamics of $E_{11}$ excitons in nanotubes on glass at sub-ps temporal resolution whereas time-resolved PL monitors the population of the emissive exciton state. Observing the same single SWCNT at the same experimental conditions avoids ambiguities arising from ensemble averaging, i.e. averaging over different species (n,m) and nanotube to nanotube variations within the same species, which may hinder the development and testing of model descriptions for the excited state dynamics. Experiments on the same nanotube using the two complementary techniques allow us to reconcile pump-probe and time-resolved PL results. We can describe both PL and pump-probe transients with the same model function and parameters that combine ultrafast EEA with exponential exciton decay[35]. The observed exponential decay times range from 5 ps to 48 ps and are attributed predominantly to non-radiative decay channels including defect-induced quenching as well as phonon-related decay mechanisms[36]. Upon increasing the excitation intensity and thereby reducing the average exciton-exciton distance we see ultrafast EEA reaching timescales down to 200 fs. Comparison of data for different SWCNTs reveals a substantial variation of the diffusional time associated with EEA presumably due to different exciton diffusion coefficients and/ or exciton localization.

## Results

### Transient interferometric scattering microscopy of SWCNTs

In this study, single SWCNTs were investigated using time-resolved absorption and TCSPC PL detection. The setup is based on a confocal microscope with high numerical aperture objective (NA = 1.49) (Fig. 1a). High NA confocal detection serves multiple purposes: First, it allows for high-resolution TA imaging limited by diffraction to ~300 nm (FWHM) at 1000 nm probe wavelength. Second, it provides efficient suppression of out-of-focus contributions and third, it maximizes the achieved absorption contrast which scales with the ratio $\sigma/A_f$, between absorption cross-section $\sigma$ and focal area $A_f$.

In the following, we briefly discuss the signal formation process in transient interferometric scattering microscopy using the setup described in Fig. 1a. The electric field at the photodiode detector results from the sum of the probe field reflected at the glass-air interface $E_{refl}$ and the probe field scattered by the SWCNT $E_{scat}$. With $I_0$ being the input probe intensity and $r$ and $s$ the corresponding reflection and scattering coefficients, respectively, the detected intensity scales as

$$I_{det} \propto I_0 \left(|r|^2 + |s|^2 + 2|r||s|\cos(\Delta\phi)\right) \quad (1)$$

For single nanoobjects, the magnitude of the scattered field is small compared to that reflected by the interface. We can thus neglect the scattered light $|s|^2$ in comparison to the reflected light $|r|^2$. We note that it is possible to detect the scattered light from medium to large-diameter nanotubes by implementing a dark field detection scheme which suppresses $|r|^2$. This can be achieved for example in the case of freely suspended SWCNTs in the absence of a reflecting interface or using cross-polarized detection[37–39]. For thin SWCNTs with diameters

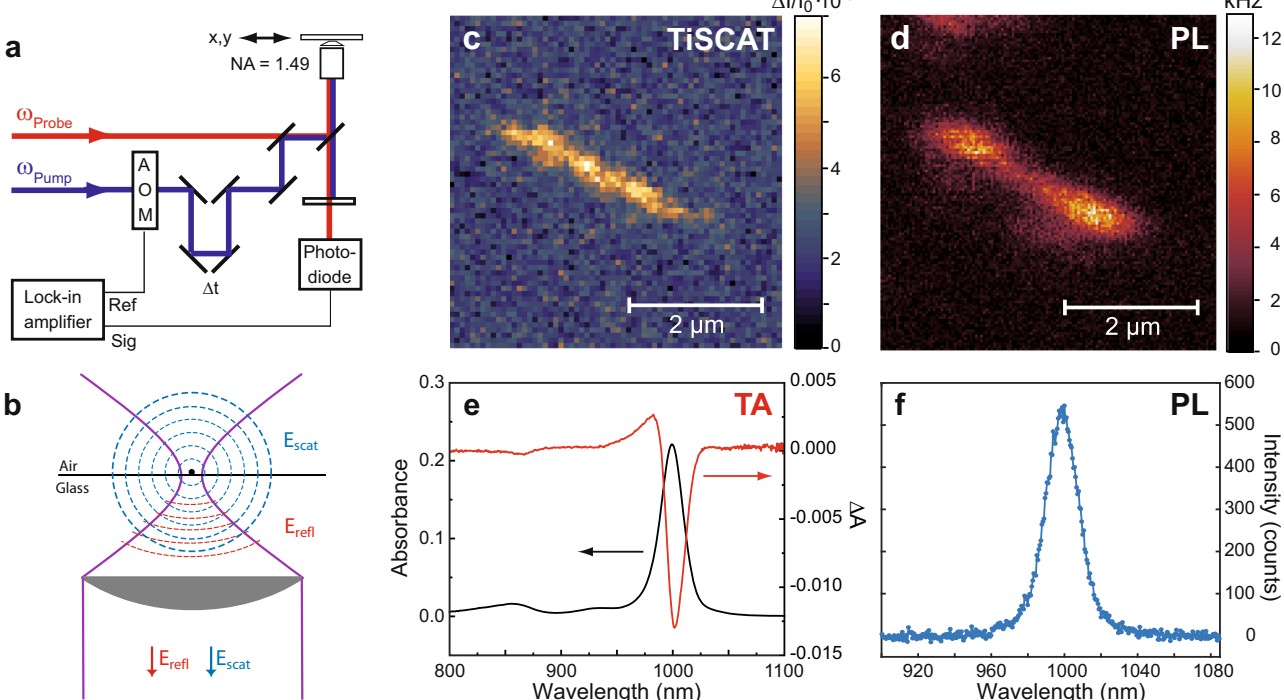

**Fig. 1 | Transient interferometric scattering microscopy of single SWCNTs.**
**a** Schematic of confocal microscopy setup for the sensitive detection of transient probe intensity changes due to few created excitons per excitation pulse. **b** In transient interferometric scattering, the detected intensity results from the interference between the field reflected by the glass substrate ($E_{refl}$) and the field scattered by the SWCNT ($E_{scat}$). Confocal transient interferometric scattering **c** and PL image **d** of a single (6,5) nanotube on glass. For both images, the nanotube was excited at 880 nm and $2.3 \times 10^5$ photons per pulse. The transient interferometric scattering image was detected at zero delay between pump and probe pulse ($\Delta t = 0$) at a probing wavelength of 1000 nm with $6.5 \times 10^4$ per pulse. **e** Transient absorption spectrum of (6,5) SWCNTs in solution (temporal average $\Delta t = 0.3-0.5$ ps) together with the ground state absorption spectrum of the sample. **f** PL spectrum of the (6,5) nanotube from **c** and **d**.

below 0.8 nm the $|s|^2$ signal intensities are not sufficient for high-bandwidth pump modulation needed to extract pump-induced signal variations with high signal-to-noise ratio.

In the case of focused excitation and detection the phase term $\Delta\phi$ in Eq. (1) contains a geometrical contribution in addition to the phase shift resulting from the scattering response of the nanoobject $\Delta\phi = \phi_{scat} + \phi_{geom}$. In focus, this geometrical phase term $\phi_{geom}$, the so-called Gouy-phase, becomes $\pi/2$. The interference term in Eq. (1) thus becomes proportional to the imaginary part of the scattering coefficient $Im(s)$, which corresponds to the absorption cross-section $\sigma$ of the nanoobject.

$$I_{int} \propto -I_0|s|\sin(\phi_{scat}) \propto -I_0 Im(s) \propto -I_0\sigma \qquad (2)$$

Relation (2) forms the basis of interferometric scattering microscopy[40] and allows us to connect the detected lock-in signal to the pump-induced changes in the sample absorption: $\Delta I/I_0 \propto -(\sigma^* - \sigma)$.

The detection of small-diameter SWCNTs such as (6,5) is particularly challenging because of the approximately linear diameter scaling of the absorption cross-section. In the present experiment on single nanotubes and very tight focusing only low probe powers in the range of few μW can be employed to avoid photodegradation. Within the range of applied probe powers, the transient interferometric scattering signal was seen to be constant (Supplementary Fig. 3). Also, linear signal scaling was observed for the used pump powers (Supplementary Fig. 3).

In Fig. 1c, we present the confocal transient interferometric scattering image of a single (6,5) SWCNT detected at zero delay time between the pump pulse at 880 nm and the probe pulse at 1000 nm together with its confocal PL image in Fig. 1d observed for the same excitation conditions. The nanotube with a length of ~2.2 μm is clearly

visible in both images. Remarkably, the nanotube cannot be detected in the simultaneously recorded elastic scattering image due to the dominating background intensity contribution resulting from the reflection at the air–glass interface (Supplementary Fig. 2). Lock-in detection at the pump modulation frequency thus serves two purposes. First, the identification of pump-induced changes of the nanotube's optical response and second, the suppression of the background light.

In the transient interferometric scattering image, the nanotube renders a positive signal corresponding to a transient reduction of absorption. This is in agreement with the observation of ground-state bleaching (GSB) in the transient absorption spectrum of (6,5) nanotubes in solution (Fig. 1e). The observed signal contrast can thus be assigned to transient ground-state bleaching or Pauli blocking caused by the creation of $E_{11}$ excitons. In addition to ground state bleaching, an additional positive signal contribution similar in magnitude can be expected to arise from stimulated emission (STE) of $E_{11}$ excitons[41]. Because of the small Stokes shift in semiconducting SWCNTs, which is on the order of few meV, no clear spectral distinction between STE emission and GSB is possible. Transient interferometric scattering and PL images with similar contrast were also obtained for single (6,4) SWCNTs on glass (Supplementary Fig. 5) confirming the general applicability of the present experimental approach. In these experiments, the probe wavelength was tuned to 880 nm matching the $E_{11}$ resonance of (6,4) SWCNTs.

## Ultrafast spectroscopy of excitons in single SWCNTs
We explored the exciton decay dynamics by recording both pump-probe and PL signal transients for a total of 18 (6,5) SWCNTs. Figure 2 displays representative pump-probe transients for two different nanotubes in a short (a) and a long time interval (b), respectively. The

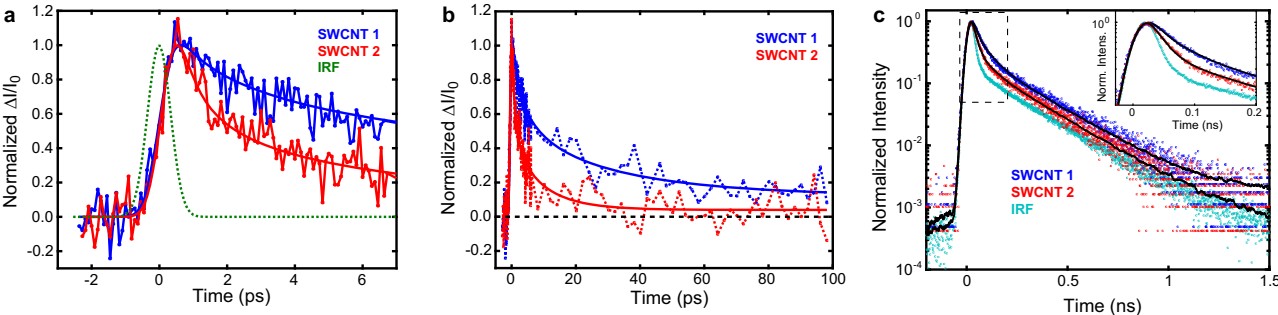

**Fig. 2 | Exciton dynamics in single SWCNTs.** Pump-probe transients detected for two different (6,5) SWCNTs in a short (**a**) and a long (**b**) time range, respectively, together with the corresponding fit functions (solid lines). **c** PL transients of the same SWCNTs.

corresponding PL transients are shown in Fig. 2c. The transients clearly reveal that the exciton decay times vary from nanotube to nanotube. In general, this can be explained by a number of factors such as different defect densities and nanotube lengths[7,8,10,42]. In addition, the ultrafast dynamics could be influenced by EEA and different EEA efficiencies as we will show below.

While the PL transients could be modeled by mono-exponential model functions in the first 1.5 ns in which the relative contribution of the long-lived dark exciton state remains small[7,8,42], this is clearly not the case for the pump-probe transients. Moreover, the decay times from mono-exponential fits of the PL transients would exhibit a strong dependence on the excitation intensity as discussed below.

To reconcile the results obtained by the two techniques a different model is needed to describe the early decay dynamics for all implemented pump energies. The fits included in Fig. 2 were obtained using the analytical model function shown in Eq. (3) developed by Srivastava and Kono in ref. 35 that accounts for EEA in addition to single exciton decay (Supplementary Note 5). Here, an ultrafast non-exponential decay time $\tau_D$ represents the time-scale of exciton diffusion finally resulting in EEA. In this description $\tau_D$ scales with the exciton diffusion coefficient $D$ and the mean exciton-exciton distance $d_0$ generated upon pulsed excitation according to $\tau_D = d_0^2/D$. The sum of radiative and non-radiative decay rates of individual excitons $\gamma$ is described by an exponential decay time $\tau_{exp} = 1/\gamma$. The ratio of the decay times is represented by $\nu = \tau_D/\tau_{exp}$[35].

$$n_{exc}(t) = \frac{\exp(-\gamma t)}{1-\nu}\left[\exp\left(\frac{1-\nu}{\nu}\gamma t\right)\cdot \mathrm{erfc}\left(\sqrt{\gamma t/\nu}\right) + \sqrt{\nu}\,\mathrm{erf}\left(\sqrt{\gamma t}\right) - \nu\right]$$

(3)

For both transient interferometric scattering and PL data, the fitted model curves were convoluted with the respective instrument response functions (IRF) also shown in Fig. 2a, c. All recorded pairs of pump-probe and PL transients can be well-described by the fit function given in Eq. (3). The observed exponential decay times, which range from 5 to 48 ps with an average of 18 ps, as well as their distribution is in general agreement with previous reports on small diameter SWCNTs[8,42]. Supplementary Fig. 7 summarizes the obtained exciton lifetimes. With PL quantum yields typically below 10 % exciton decay in semiconducting nanotubes is dominated by non-radiative decay processes[43] which can include multi-phonon decay, phonon-assisted indirect exciton ionization[36] as well as quenching at defect sites and nanotube ends[44,45]. In the present experiments, TiSCAT and PL transients were recorded at the same position in the center of the investigated SWCNTs for direct comparability. No systematic influence of length-dependent end quenching on the exponential decay time was observed because of a minimum nanotube length of 500 nm exceeding the exciton diffusion length of about 100 nm[46] (Supplementary Fig. 8).

We note that for delay times between 10 and 100 ps the pump-probe transients could also be described by the $t^{-1/2}$ decay reported in TA literature attributed to EEA[19,25]. This decay function, however, cannot describe the initial time dependence. Moreover, on longer timescales the dynamics of $E_{11}$ excitons is precisely monitored over several orders of magnitude by TCSPC and described by exponential functions (Fig. 2)[7,8,10,42,47]. We further note that the fit function given by Eq. (3) can also be used to describe the transients recorded for (6,4) SWCNT on glass (Supplementary Fig. 6) indicating the general suitability of the model.

## Ultrafast exciton–exciton annihilation in single SWCNTs

The experiment described here, in combination with the parameters obtained from the model function in Eq. (3), now allows us to investigate the EEA dynamics in further detail and on the single nanotube level. This is done by comparing the measured temporal evolution of the exciton number with predictions from the diffusive EEA model. The early decay dynamics of a single (6,5) SWCNT is shown in Fig. 3a for two excitation powers. Significantly faster decay is observed for higher excitation power. Figure 3b illustrates the dependence of the derived diffusion time $\tau_D$ on the pump intensity (bottom axis) for a set of different SWCNTs. In this analysis, the exponential decay time, which is obtained from Eq. (3) for each SWCNT, remains constant as expected for the decay of individual isolated excitons.

As seen in Fig. 3b, experimental $\tau_D$ values drop to only 200 fs when the exciton number in the laser focus and thus the exciton density increase, consistent with ultrafast exciton-exciton annihilation. This timescale appears to be in general agreement with those reported for reaction-limited EEA in the regime of extremely high exciton densities at nearly complete saturation of the TA signals for SWCNTs ensembles of different chiralities in solution[25]. The highest excitation densities used in the present experiments, however, are about 100 times lower with linear power scaling of the TA signal (Supplementary Fig. 3a). Variations in $\tau_D$ as seen in Fig. 3b could indicate different effective exciton diffusion coefficients $D$ as discussed below.

In the following, we assess how well diffusion times obtained from the fit of experimental data with Eq. (3) are consistent with experimentally reported diffusion coefficients. To do so we calculate the diffusion times $\tau_D = d_0^2/D$ from the mean distance of excitons generated upon pulsed excitation $d_0$ in our experiment and literature values of the diffusion coefficient $D$. The number of incident photons per pulse $N_{photons}$ can be connected to the number of initially created excitons $N_{excitons}$ using the absorption cross section $\sigma$ of (6,5) SWCNTs[48] according to $N_{excitons} = \sigma N_{photons}$ (Fig. 3b) (Supplementary Note 7). From this we can calculate the average exciton–exciton distance $d_0 = d_{fl}/(2N_{excitons})$[35] considering a Gaussian spatial distribution of excitons following the profile of the focused excitation pulse. The number of excitons specified on the upper $x$-axis in Fig. 3b corresponds to the total number of initially created excitons within the

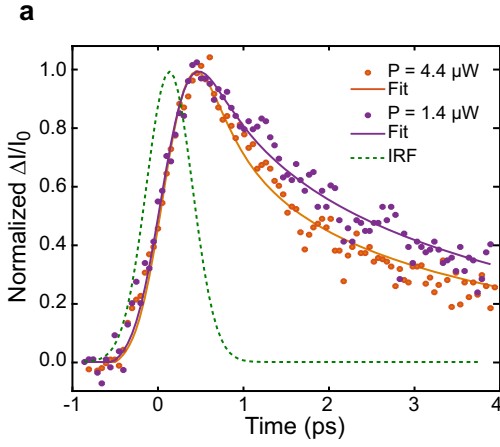

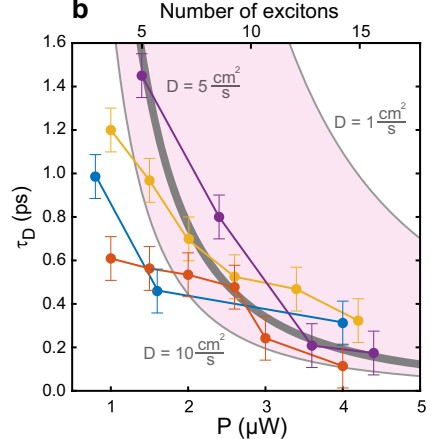

**Fig. 3 | Ultrafast exciton-exciton annihilation in single SWCNTs. a** Pump-probe transient detected for a single (6,5) SWCNT for pump powers of 4.4 μW and 1.4 μW together with the corresponding fitted model curves, respectively. **b** Pump power dependence of the diffusion time $\tau_D$ for 4 different nanotubes derived from fitting the detected transients. The calculated diffusional times for a diffusion coefficient of $D = 5 \, cm^2/s$ are shown as solid gray line.

diffraction-limited illuminated nanotube segment length of $\approx \lambda/NA = 880nm/1.49 \approx 600nm$.

The exciton diffusion coefficient of freely suspended thin semi-conducting nanotubes has recently been reported for both bright and dark excitons[7]. Using the equation for the diameter-dependent diffusion coefficient reported in ref. [7] for 0.747 nm, the diameter of (6,5) SWCNTs, results in an estimated value of $D = 7 \, cm^2/s$. Because of coupling to the heterogeneous polymer environment, this value probably represents an upper limit. Indeed, for micelle-encapsulated and polymer-wrapped SWCNTs on substrates values ranging from $D = 0.4$ to $7.5 \, cm^2/s$ have been reported[9,46,49].

Using $D = 5 \, cm^2/s$ and the initial exciton densities determined above the calculated diffusional time $\tau_D = d_0^2/D$ and its excitation intensity dependence agree very well with the experimental data recorded for single (6,5) SWCNTs (Fig. 3b). We note however that for low excitation powers the predicted times appear to be systematically longer than the experimental ones, whereas a better match is seen for high powers. We speculate that the observed deviation of observed diffusion times might be due to spatially heterogeneous diffusion coefficients as expected for SWCNT environments that are known to lead to efficient exciton localization[50,51].

## Finite exciton numbers and localization

To explore the influence of exciton localization on the EEA dynamics we performed Monte-Carlo simulations of exciton diffusion and localization using the parameters of the single nanotube experiments (Supplementary Note 8). Figure 4 compares the result of the Monte-Carlo simulation with the analytical curve for the same diffusion coefficient $D = 5 \, cm^2/s$. Whereas the quantized and analytical curves agree for high exciton numbers the Monte-Carlo simulations lead to longer diffusional times for smaller exciton numbers. This can be understood by the fact that for small exciton numbers the relative contribution of single excitons that remain without annihilation partner, becomes more relevant, as has also been noted in ref. [35].

Remarkably, introducing a single exciton localization site in the center of the illuminated section of the SWCNT significantly speeds up EEA for excitation powers below 3 μW, equivalent to about 10 initially created excitons (Fig. 4). This is surprising because in diffusion-limited reaction systems inhibiting the motion of one reaction partner is expected to slow down the diffusion time corresponding to a reduced diffusion coefficient from $2 \cdot D$ to $D$. For a spatially confined excitation region, however, the probability distribution of excitons is strongly heterogeneous. Localizing excitons

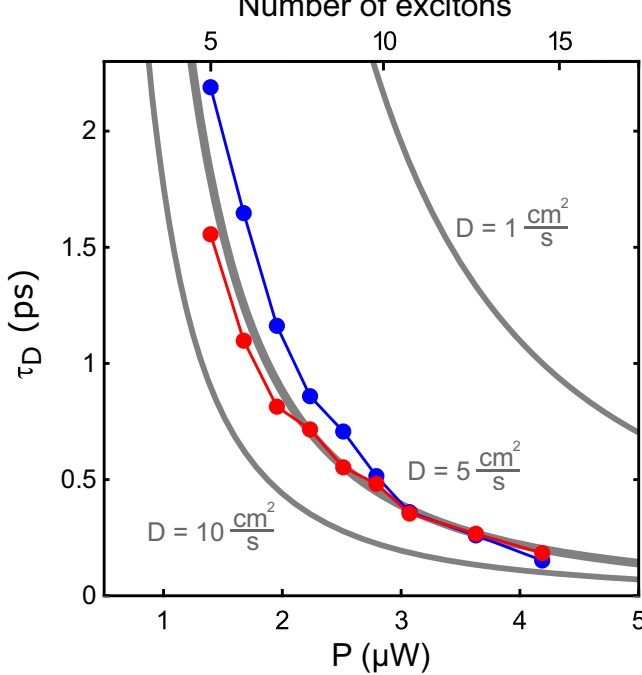

**Fig. 4 | Exciton−exciton annihilation for finite exciton numbers and localization.** Monte-Carlo simulations of the diffusion times with (red points) and without (blue points) exciton localization and a diffusion coefficient of $D = 5 \, cm^2/s$. Analytically calculated diffusion times for $D = 1$, 5 and 10 $cm^2/s$ are shown as solid gray lines.

thus prevents them from escaping the high exciton density region, thereby sustaining a high probability for exciton-exciton annihilation encounters, higher than if all excitons were free to roam around. For high excitation densities in which EEA becomes extremely fast this effect is less pronounced (Fig. 4). As a result, the diffusional times are seen to decrease more slowly upon increasing the excitation power in the presence of localization, closely matching the scaling behavior observed for the experimental data in Fig. 3b. Adding a second localization site, on the other hand, has a smaller effect. The actual impact of exciton localization on the timescale of EEA will depend on the number and distribution of localization site(s) within the illuminated nanotube segment and we tentatively attribute variations in $\tau_D$ from one nanotube to the next to such heterogeneities.

In summary, we investigated the exciton decay dynamics in single semiconducting (6,5) single-walled carbon nanotubes using transient absorption and time-resolved PL microscopy. The combined data sets obtained for individual nanotubes allowed us to formulate and test a unified description for both types of transients which combines exciton-exciton annihilation described by a diffusional time constant and single exciton relaxation governed by exponential decay. Pump-probe transients detected when increasing the average number of excitons created by pulsed excitation from about 2 to 15 reveal ultrafast annihilation of excitons on a time scale reaching down to 200 fs. The general power dependence of the diffusional times can be very well described using the initial average exciton-exciton distance and reported values of the diffusion coefficient of (6,5) SWCNTs. Measurements on different nanotubes reveal a distribution of diffusional times. Monte-Carlo simulations of the time-dependent exciton population show that significantly faster EEA can be caused by a single localization site within the initially excited section of the nanotube. We suggest that different numbers and positions of localization sites within the probed nanotube section are the cause for the observed variation of diffusional times from nanotube to nanotube. The reported results provide a unified description of the exciton dynamics in semiconducting SWCNTs observed by time-resolved PL and transient absorption spectroscopy. The developed experimental platform and understanding of the exciton dynamics provide the basis for investigations of energy transfer processes in hybrid systems formed for example by dye-filled or covered SWCNTs[52–54].

## Methods

### Polymer-wrapped (6,5) SWCNTs

The SWCNT raw material, synthesized by the CoMoCAT procedure, was purchased from Sigma Aldrich (SG65). Organic polymer stabilized SWCNT samples were produced from shear-mixing the SWCNT raw material and poly[(9,9-dioctylfluorenyl-2,7-diyl)-alt-(6,6′-{2,2′-bipyridine})] copolymer in toluene for 13 h followed by filtration. 30 µl of this suspension was deposited on thin glass cover slides by spin-coating. To reduce environmental perturbations and to decouple the SWCNTs from the glass surface, cover slides were precoated by a thin (30 nm) polystyrene layer.

### TiSCAT and time-resolved PL microscopy

TiSCAT and time-resolved PL measurements were performed on the confocal microscope setup illustrated in Fig. 1a and in Supplementary Fig. 1. Residual pump laser light was blocked by a spectral long-pass filter. In TiSCAT, the pump pulse intensity was modulated at 96 kHz using an acousto optic modulator. The probe pulse intensity was detected at the modulation frequency by a sensitive photodiode in combination with a lock-in amplifier. For PL measurements, the probe pulse was blocked and the emitted light was detected by a fast avalanche photodiode connected to time-correlated single photon counting electronics. The IRFs in case of TiSCAT and time-resolved PL detection are included in Fig. 2a,c.

## Data availability

The data generated in this study have been deposited in the figshare database under accession code https://doi.org/10.6084/m9.figshare.21326046.

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

## Acknowledgements

K.B. and A.H. acknowledge financial support from the Deutsche Forschungsgemeinschaft (DFG) through Germany's Excellence Strategy-EXC 2089/1-390776260. We thank Ivonne Vollert, Julius-Maximilians-University Wuerzburg, for the preparation of polymer-dispersed nanotube suspensions.

## Author contributions

K.B. and A.H. conceived and designed the experiment. K.B. carried out the experiments and the numerical simulations and analyzed the data. K.B. and A.H. developed the model with support from T.H. T.H. provided the nanotube material. All authors wrote and reviewed the manuscript.

## Funding

## Competing interests
The authors declare no competing interests.
