## [Peer Review File · Nature Communications]

Probing the Ultrafast Dynamics of Excitons in Single Semiconducting Carbon NanotubesREVIEWER COMMENTS

Reviewer #1 (Remarks to the Author):

Review comments for

Manuscript ID: NCOMMS-22-14484

Title: Probing the Ultrafast Dynamics of Excitons in Single Semiconducting Carbon Nanotubes

Konrad Birkmeier, Tobias Hertel, and Achim Hartschuh

Submitted to: Nature Communications

Comments:

The submitted manuscript claims to develop a new measurement platform to detect exciton transport in SWCNTs using a combination of transient interferometric scattering (TiSCAT) and transient photoluminescence (TRPL) measurements. Authors claim this approach would be more effective to evaluate ultra-fast exciton transport phenomena in SWCNTs compared to the existing approach (using transient absorption, TA & TRPL).

However, the results/outcome sections do not reflect the above claim clearly. Therefore, the importance of the work is not clear. The new scientific findings, that can be only observed utilizing the mentioned transient interferometric scattering (TiSCAT) as claimed by this manuscript, have not been demonstrated clearly in the experimental results section. Hence the claimed novelty of this work can not be cross-verified.

Moreover, the main outcome of the results suggested the exciton-exciton annihilation process in SWCNT. However, this is well reported previously. Therefore, the novelty of this approach is not clear.

In summary, the current version of the manuscript is premature and far from a publishable quality adequate for a publication. While this format may be sufficient for a specific journal, however, may not be suitable for Nature Communications. The current version of the manuscript is not appropriate for publication and should be recommended as a "rejection" in its present form.

The issues listed below are critical for the publication and need consideration to justify all manuscript claims, logical justifications, and clarity.

1) The importance of the work is not clear. For example, on page 4; line 77, the authors mentioned: "significant discrepancies between the models derived from time-resolved PL and transient absorption studies persist and a unifying picture simultaneously describing data from both spectroscopic probes by a single comprehensive mechanism appears to be lacking."

However, the manuscript's focus especially the "title" and/or abstract do not reflect a new scientific finding/mechanism that was not reported in the current literature. If there is a "new single comprehensive mechanism" that was obtained in this study, that "new single comprehensive mechanism" should have revealed previously unreported scientific findings. The manuscript's focus and title/abstract should report this new scientific finding instead. Currently, the manuscript's focus is quite broad and general, hence the importance of the work could not be determined.

2) There are numerous reports in the literature (for example, Phys. Rev. B 2005, 72, 195415; Phys. Rev. Lett. 2005, 94, 207401; Nat. Phys. 2008, 5, 54; etc.) demonstrating very fast (fs to ps) exciton transport in SWCNT. Then what is the new information that can be only observed utilizing the mentioned transient interferometric scattering (TiSCAT) as claimed by this manuscript? The novelty of TiSCAT that is claimed in the introduction part, has not been demonstrated clearly in the experimental results section.

3) Moreover, the main outcome of the results suggested the exciton-exciton annihilation

process in SWCNT. However, this is well reported previously (for example, Nanoscale, 2016, 8, 1618; Phys. Rev. B 73, 115432). Then what is the new phenomenon that is observed here and not published earlier? In other words, then the novelty of the outcomes is not clear here.

4) Furthermore, the authors mentioned in the conclusion section, page 13, line 260: "The combined data sets obtained for individual nanotubes allowed us to formulate and test a unified description for both types of transients which combines exciton-exciton annihilation described by a diffusional time constant and single exciton relaxation governed by exponential decay."

These observations were well reported in previous reports as mentioned above. Therefore, what is the new phenomenon that is observed here and not published earlier, is not clear and crucially important to justify the acceptability of this work.

Reviewer #2 (Remarks to the Author):

The manuscript by K. Birkmeier et al. reports an iSCAT-based study of the excited state dynamics of single-walled carbon nanotube, particularly focusing on (6,5) nanotubes in the present work. It is interesting that both PL and pump-probe spectroscopy are combined to offer insights into the exciton diffusion and exciton-exciton annihilation processes in these materials. However, there are several important issues that need to be thoroughly and carefully addressed, before the manuscript can be considered for publication.

(1) When it is stated that the nanotube cannot be detected in the elastic scattering image, the definition of elastic scattering image here is not clear and needs to be included. This will allow for a better understanding of the iSCAT signal and its strength. In addition, is it possible to detect the CNT with just the probe laser (i.e, the pump being turned off) and without using the lock-in amplifier?

(2) More details on the experimental setup should be provided. For example, it may be nontrivial that the probe laser is right at the resonance of the (6,5) nanotube. Was it based on a supercontinuum laser? In addition, are the pump and probe pulses focused on the same spot of the sample, or if their position can be independently varied, which would yield information on the exciton diffusion within the nanotubes.

(3) For the pump-probe transients in Fig. 2(a) and 2(b), are the plotted data based on a single pixel within the nanotube, or obtained by integrating over the entire area of the nanotube? An advantage of the technique presented here is that the transients can be obtained from all the pixels. While the authors clearly show that there are nanotube-to-nanotube variations in the dynamics, are there intratube variations (for example, if the tube ends decay faster than the middle of the tube)?

(4) For estimating the number of initially created excitons using the cross-section value obtained from literature, can saturable absorption be relevant here in the comparatively high excitation intensity used in the pump-probe experiments? In addition, it appears from Fig. 1d that there are brighter emission at the two ends of the nanotubes; some comments on this would be helpful: for example, is this a universal effect observed on all the examined nanotubes?

(5) More details on the Monte Carlo simulation should be provided.

(6) While it is quite useful to combine pump-probe results and time-resolved PL data to offer a unifying picture of excited state dynamics of (6,5) carbon nanotubes, how are the two data sets (Fig. 2) with very different timescales connected? Is the first 0.2 ns of the PL data used primarily for comparison with the pump-probe results, and how reliable is it (some

description on the steps taken to remove the IRF would be useful).

Reviewer #3 (Remarks to the Author):

The manuscript entitled "Probing the ultrafast dynamics of excitons in single semiconducting carbon nanotubes" by K. Birkmeier et al. describes the dynamics of excitons in a (6,5) single-walled carbon nanotube using time-resolved interferometric scattering spectroscopy and photoemission spectroscopy. So many studies on ultrafast dynamics of excitons in carbon nanotubes have already been reported. Nevertheless, the main advantage of this study is that the authors measured the dynamics of excitons in a single carbon nanotube, which is a non-trivial work. Most of the previous works describe the dynamics of exciton of carbon nanotubes in in bulk state. Furthermore, they comprehensively investigated the exciton dynamics in single-wall carbon nanotubes, and they also adequately cited the previous studies for creating this manuscript. Therefore, the manuscript is of interest to nanotechnology or material science communities.

Here, I would like to raise several remarks, which may improve the quality of the manuscript.

1. They addressed that they measured 18 carbon nanotubes (page 8) and investigated 30 decay constants (Supplementary Figure 3). They also addressed that the exciton decay times vary from nanotube to nanotube because of defect density and nanotube length. However, they can estimate the nanotube length using their technique as shown in Fig. 1c. Therefore, at least, they can statistically understand the changes in exciton decay time affected by the nanotube length. Figure 3b with the information on nanotube length is essential for the further understanding of the dynamics of the excitons.
2. Related to 1, regarding the defect density, is it possible to derive the information about the defect density from the E11* or E11*- peaks from absorption or PL spectra? Is there any correlation?
3. They used two time-resolved spectroscopic measurements onto the (6,5) single-wall carbon nanotubes. They used time-resolved interferometric scattering spectroscopy in the early timescale (1 – 100 ps) and used time-resolved PL spectroscopy in the later timescale (0.1 – 1 ns). However, these two measurements do not temporally overlap together. For example, in Fig. 2b, the transients do not fully relax at 100 ps. Does this very slow decay correspond to the decay observed by time-resolved PL spectroscopy? If their setup allows them to measure extended time (~1 ns or ~30 cm), it is interesting to compare the signals of time-resolved interferometric scattering spectroscopy and photoemission spectroscopy.
4. Is their discussion of exciton dynamics in a semiconducting nanotube valid for other carbon nanotubes with different chirality?
5. In the abstract, they stated the segment length of the carbon nanotubes is 600 nm; however, in the main text, there is no information about the segment length.
6. Did the authors synthesize the (6,5) single-walled carbon nanotube? If yes, it is better to show the synthesis procedure shortly or cite the article that shows the synthesis procedure. If they bought the sample, it is better to show the provider. They only showed how to wrap the carbon nanotubes with the polymer.

7. (minor) They addressed the number of excitons as 2–15 in the abstract and 2–16 in summary. Please correct it.

Review comments for

Manuscript ID: NCOMMS-22-14484

Title: Probing the Ultrafast Dynamics of Excitons in Single Semiconducting Carbon Nanotubes

Konrad Birkmeier, Tobias Hertel, and Achim Hartschuh

Submitted to: Nature Communications

Comments:

The submitted manuscript claims to develop a new measurement platform to detect exciton transport in SWCNTs using a combination of transient interferometric scattering (TiSCAT) and transient photoluminescence (TRPL) measurements. Authors claim this approach would be more effective to evaluate ultra-fast exciton transport phenomena in SWCNTs compared to the existing approach (using transient absorption, TA & TRPL).

However, the results/outcome sections do not reflect the above claim clearly. Therefore, the importance of the work is not clear. The new scientific findings, that can be only observed utilizing the mentioned transient interferometric scattering (TiSCAT) as claimed by this manuscript, have not been demonstrated clearly in the experimental results section. Hence the claimed novelty of this work can not be cross-verified.

Moreover, the main outcome of the results suggested the exciton-exciton annihilation process in SWCNT. However, this is well reported previously. Therefore, the novelty of this approach is not clear.

In summary, the current version of the manuscript is premature and far from a publishable quality adequate for a publication. While this format may be sufficient for a specific journal, however, may not be suitable for Nature Communications. The current version of the manuscript is not appropriate for publication and should be recommended as a “*rejection*” in its present form.

The issues listed below are critical for the publication and need consideration to justify all manuscript claims, logical justifications, and clarity.

- 1) The importance of the work is not clear. For example, on page 4; line 77, the authors mentioned: “significant discrepancies between the models derived from time-resolved PL and transient absorption studies persist and a unifying picture simultaneously

describing data from both spectroscopic probes by a single **comprehensive mechanism appears to be lacking.**”

However, the manuscript's focus especially the “title” and/or abstract do not reflect a new scientific finding/mechanism that was not reported in the current literature. If there is a “new single comprehensive mechanism” that was obtained in this study, that “new single comprehensive mechanism” should have revealed previously unreported scientific findings. The manuscript’s focus and title/abstract should report this new scientific finding instead. Currently, the manuscript’s focus is quite broad and general, hence the importance of the work could not be determined.

- 2) There are numerous reports in the literature (for example, Phys. Rev. B 2005, 72, 195415; Phys. Rev. Lett. 2005, 94, 207401; Nat. Phys. 2008, 5, 54; etc.) demonstrating very fast (fs to ps) exciton transport in SWCNT. Then what is the new information that can be only observed utilizing the mentioned transient interferometric scattering (TiSCAT) as claimed by this manuscript? The novelty of TiSCAT that is claimed in the introduction part, has not been demonstrated clearly in the experimental results section.
- 3) Moreover, the main outcome of the results suggested the exciton-exciton annihilation process in SWCNT. However, this is well reported previously (for example, Nanoscale, 2016, 8, 1618; Phys. Rev. B 73, 115432). Then what is the new phenomenon that is observed here and not published earlier? In other words, then the novelty of the outcomes is not clear here.
- 4) Furthermore, the authors mentioned in the conclusion section, page 13, line 260: “The combined data sets obtained for individual nanotubes allowed us to formulate and test a unified description for both types of transients which combines **exciton-exciton annihilation described by a diffusional time constant** and single exciton relaxation governed by exponential decay.”

These observations were well reported in previous reports as mentioned above. Therefore, what is the new phenomenon that is observed here and not published earlier, is not clear and crucially important to justify the acceptability of this work.

Reviewer #1 (Remarks to the Author):**Comments:**

The submitted manuscript claims to develop a new measurement platform to detect exciton transport in SWCNTs using a combination of transient interferometric scattering (TiSCAT) and transient photoluminescence (TRPL) measurements. Authors claim this approach would be more effective to evaluate ultra-fast exciton transport phenomena in SWCNTs compared to the existing approach (using transient absorption, TA & TRPL). However, the results/outcome sections do not reflect the above claim clearly. Therefore, the importance of the work is not clear. The new scientific findings, that can be only observed utilizing the mentioned transient interferometric scattering (TiSCAT) as claimed by this manuscript, have not been demonstrated clearly in the experimental results section. Hence the claimed novelty of this work can not be cross-verified. Moreover, the main outcome of the results suggested the exciton-exciton annihilation process in SWCNT. However, this is well reported previously. Therefore, the novelty of this approach is not clear. In summary, the current version of the manuscript is premature and far from a publishable quality adequate for a publication. While this format may be sufficient for a specific journal, however, may not be suitable for Nature Communications. The current version of the manuscript is not appropriate for publication and should be recommended as a "rejection" in its present form. The issues listed below are critical for the publication and need consideration to justify all manuscript claims, logical justifications, and clarity.

1) The importance of the work is not clear. For example, on page 4; line 77, the authors mentioned: "significant discrepancies between the models derived from time-resolved PL and transient absorption studies persist and a unifying picture simultaneously describing data from both spectroscopic probes by a single comprehensive mechanism appears to be lacking."

However, the manuscript's focus especially the "title" and/or abstract do not reflect a new scientific finding/mechanism that was not reported in the current literature. If there is a "new single comprehensive mechanism" that was obtained in this study, that "new single comprehensive mechanism" should have revealed previously unreported scientific findings. The manuscript's focus and title/abstract should report this new scientific finding instead. Currently, the manuscript's focus is quite broad and general, hence the importance of the work could not be determined.

Reply: In our manuscript, we did not claim to have obtained a new single comprehensive mechanism. Instead, we stated that we formulated a unified model combining unimolecular decay and exciton-exciton annihilation. Here "unified" refers to the

fact that the model can explain transients obtained by both pump-probe and time-resolved PL measurements. We agree with the reviewer in that the statement on page 4; line 77 "...a unifying picture simultaneously describing data from both spectroscopic probes by a single comprehensive *mechanism* appears to be lacking" could be misleading and changed the wording to "...single comprehensive *model based on the same transition rates* appears to be lacking". In addition, we included examples of literature reports on pump-probe and time-resolved PL that illustrate the above-mentioned discrepancies and missing connections between the two.

Added text: "For example, whereas the signal transients obtained by transient absorption and time-resolved PL have been both explained by three level systems including optically dark states, very different branching and relaxation rates have been derived (refs. 7, 8, 18) possibly reflecting the different timescales probed by these experiments. Other reports using transient absorption spectroscopy found tri-exponential decay dynamics on the sub ps to few tens of ps timescale from ensemble experiments in solution, which were attributed to phonon-assisted processes (ref. 31) with no direct counterpart in PL experiments".

2) *There are numerous reports in the literature (for example, Phys. Rev. B 2005, 72, 195415; Phys. Rev. Lett. 2005, 94, 207401; Nat. Phys. 2008, 5, 54; etc.) demonstrating very fast (fs to ps) exciton transport in SWCNT. Then what is the new information that can be only observed utilizing the mentioned transient interferometric scattering (TiSCAT) as claimed by this manuscript? The novelty of TiSCAT that is claimed in the introduction part, has not been demonstrated clearly in the experimental results section.*

Reply: We agree with the reviewer in that very fast exciton-exciton annihilation (EEA) in SWCNTs, which indicates fast transport, has been reported before. In our manuscript we have cited the corresponding literature on EEA starting with the first reports in 2004 to the more recent literature in 2016 (refs. 23-26). To illustrate this further, we included three additional references on this topic (Phys. Rev. Lett. 96, 057407 (2006), J. Phys. Chem. C 117, 1974 (2013), Nanoscale 11, 14907 (2019)). The new information in our manuscript is that exciton-exciton annihilation and the underlying diffusive transport can be observed in individual semiconducting SWCNTs in real-time and that the diffusional time can be predicted from the number of incident photons and the known diffusion coefficient. Crucially, significant variations in the diffusional time from nanotube to nanotube are observed for the first time that indicate varying degrees of exciton localization. We note that although exciton-exciton annihilation has been reported in many publications as mentioned

above, conflicting interpretations still exist, such as the occurrence of a reaction limited regime for times between 0.4 and 1.2 ps after pulsed excitation (ref. 25) for which we find no indications.

3) Moreover, the main outcome of the results suggested the exciton-exciton annihilation process in SWCNT. However, this is well reported previously (for example, Nanoscale, 2016, 8, 1618; Phys. Rev. B 73, 115432). Then what is the new phenomenon that is observed here and not published earlier? In other words, then the novelty of the outcomes is not clear here.

Reply: We agree with the reviewer in that exciton-exciton annihilation has been reported by many groups before, as discussed in our answer to point 2. In our manuscript we observe this phenomenon for the first time in individual SWCNTs on the sub-ps to ps timescale. These measurements allow to correlate exciton diffusion times and initial exciton densities, the latter ranging from about 2 to 15 excitons in the illuminated segment controlled by the excitation laser power, without ambiguities resulting from averaging over different nanotube species, different orientations and different lengths (Fig. 3b). The good agreement with the calculated diffusion times $\tau_D = d_0^2/D$ confirms that besides the slower mono-exciton relaxation, no additional excited state processes need to be involved to describe the ultrafast decay dynamics in semiconducting SWCNTs. At the same time, we observe significant nanotube to nanotube variations that underline the importance of single nanotube experiments in avoiding ensemble averaging.

4) Furthermore, the authors mentioned in the conclusion section, page 13, line 260: "The combined data sets obtained for individual nanotubes allowed us to formulate and test a unified description for both types of transients which combines exciton-exciton annihilation described by a diffusional time constant and single exciton relaxation governed by exponential decay."

These observations were well reported in previous reports as mentioned above. Therefore, what is the new phenomenon that is observed here and not published earlier, is not clear and crucially important to justify the acceptability of this work.

Reply: Despite the large number of literature reports on EEA and the ultrafast excited dynamics in general, which are mostly based on transient absorption experiments, a direct connection to the decay dynamics observed in time-resolved PL experiments is missing. Indeed, whereas the signal decay in transient absorption experiments has been described in different publications by mono-, bi- and tri-expo-

ponential decay as well as $1/\sqrt{t}$ relaxation with strongly varying time constants involving ultrafast couplings to dark states or phonon-mediated processes a direct connection to the dynamics in time-resolved PL data of the same nanotube species has been elusive. We illustrated the discrepancies between the models and time-scales reported from transient absorption and time-resolved PL experiments more explicitly and by providing further examples in the revised version of the manuscript. One reason for this are certainly the different time-scales for which the two techniques are ideally suited. Another reason will be that while time-resolved PL measurements on single nanotubes have been reported many years ago, providing insight into the relaxation dynamics beyond ensemble averaging, corresponding TA measurements of thin single semiconducting nanotubes, however, were not available so far. Crucially, the data obtained by the present combination of the two techniques can be described involving only two decay mechanisms without the need of assuming additional ultrafast couplings to dark states or phonon-mediated processes. In conclusion, the phenomena we have observed are known, in principle. However, by avoiding ensemble averaging we show that the ultrafast exciton decay dynamics in semiconducting carbon nanotubes can be quantitatively explained using only two mechanisms. We have highlighted this point clearly and at several occasions in our manuscript.

Reviewer #2 (Remarks to the Author):

The manuscript by K. Birkmeier et al. reports an iSCAT-based study of the excited state dynamics of single-walled carbon nanotube, particularly focusing on (6,5) nanotubes in the present work. It is interesting that both PL and pump-probe spectroscopy are combined to offer insights into the exciton diffusion and exciton-exciton annihilation processes in these materials. However, there are several important issues that need to be thoroughly and carefully addressed, before the manuscript can be considered for publication.

(1) When it is stated that the nanotube cannot be detected in the elastic scattering image, the definition of elastic scattering image here is not clear and needs to be included. This will allow for a better understanding of the iSCAT signal and its strength. In addition, is it possible to detect the CNT with just the probe laser (i.e., the pump being turned off) and without using the lock-in amplifier?

Reply: We thank the reviewer for this suggestion. We added the simultaneously detected elastic scattering image as Supplementary Fig. 2. Besides signal fluctuations due to laser instabilities, elastic scattering at a single detection energy lacks the

specificity needed to identify SWCNTs. This means that SWCNTs could not be distinguished from other scattering objects, such as possible dust particles or residual surfactant. In the present experiment this specificity is obtained by detecting the pump-induced changes at the probe energy.

(2) More details on the experimental setup should be provided. For example, it may be nontrivial that the probe laser is right at the resonance of the (6,5) nanotube. Was it based on a supercontinuum laser? In addition, are the pump and probe pulses focused on the same spot of the sample, or if their position can be independently varied, which would yield information on the exciton diffusion within the nanotubes.

Reply: We agree with the reviewer and included more detailed information on the experimental setup in Supplementary Note 1 and Supplementary Fig. 1. Briefly, the setup combines a pulsed laser system with a scanning confocal microscope. A tunable Ti:Sa oscillator provides laser pulses with a duration of about 150 fs at a repetition rate of 76 MHz. The laser was operated at the pump wavelength of 880 nm and divided into pump and probe beams by a 50 / 50 beam splitter. The amplitude of the pump-beam was modulated using an acousto optic modulator. Probe pulses were created by pumping a photonic crystal fiber for continuum generation followed by spectral filtering using a narrow bandpass filter (bandwidth 10 nm) centered at 1000 nm. Pump- and probe beams were recombined using a beam splitter. Their lateral overlap in the focus of the microscope objective was controlled using a piezo-electric mirror positioned in a conjugate Fourier plane. While this configuration allows to systematically scan pump vs. probe focus, as suggested by the reviewer, we did not achieve clear signatures of exciton diffusion along the nanotubes yet. For the investigated nanotubes the exciton diffusion length is expected to be on the order of 100 nm (e.g. Science 316, 1465 (2007), Nano Lett. 10, 1595 (2010), Nano Lett. 12, 5091 (2012)), significantly smaller than the focal width and the spatial resolution of the experiment: Diffusion will thus lead to a smaller broadening effect only, which will be further reduced by exciton-exciton annihilation at higher excitation densities.

(3) For the pump-probe transients in Fig. 2(a) and 2(b), are the plotted data based on a single pixel within the nanotube, or obtained by integrating over the entire area of the nanotube? An advantage of the technique presented here is that the transients can be obtained from all the pixels. While the authors clearly show that there are nanotube-to-nanotube variations in the dynamics, are there intratube variations (for example, if the tube ends decay faster than the middle of the tube)?

Reply: The transients were obtained for a single position located in the center of the investigated nanotubes. The signal thus originates from a segment of the nanotube with a length of about 300 nm given by the spatial resolution of the experiment. In the experiment the spatial resolution is limited by diffraction of light and can be determined for example by measuring the width of the signal in the scan image in Fig. 1c. We agree with the reviewer in that exciton decay is expected to be faster at the ends compared to the middle of the nanotube. So far, we did not observe significant or systematic differences in the dynamics for different positions within single nanotubes. This can be attributed to the spatial averaging resulting from the finite resolution, which is significantly larger than the diffusion lengths reported for (6,5) SWCNTs on glass of about 100 nm (e.g. Science 316, 1465 (2007), Nano Lett. 10, 1595 (2010), Nano Lett. 12, 5091 (2012)). In addition, even for the lowest excitation densities used in the experiment and needed to observe a clear signal in TiSCAT we found efficient exciton-exciton annihilation leading to rapid initial decay (Fig. 3b), which can be expected to dominate with respect to end-quenching. This discussion is related to the first comment of reviewer 3.

(4) For estimating the number of initially created excitons using the cross-section value obtained from literature, can saturable absorption be relevant here in the comparatively high excitation intensity used in the pump-probe experiments? In addition, it appears from Fig. 1d that there are brighter emission at the two ends of the nanotubes; some comments on this would be helpful: for example, is this a universal effect observed on all the examined nanotubes?

Reply: We agree with the reviewer in that saturable absorption could influence the actual absorption cross-section. Excitation power dependent experiments reported in the literature indicate that the exciton correlation length in (6,5) SWCNT is on the order of 3-13 nm influenced by the dielectric environment (ref 38, *Nat. Phys.* 5, 54 (2008)). This means that for the highest exciton densities of 15 excitons per 600 nm, saturation could indeed lead to a reduction of absorption by about $(15 \cdot (3 \text{ to } 13)) / 600 = 8 - 32 \%$. However, for all excitation powers used in the experiments we observe linear scaling of the detected transient absorption signal (Supplementary Fig. 3a) giving no indications for significant saturation.

Brighter emission is not a general finding and appears to be specific for the nanotube seen in Fig. 1d. For nanotubes on substrates non-uniform PL intensities are rather common and can be attributed to inhomogeneous local environments (e.g. *ACS Nano*, 4, 5914 (2010)).

(5) *More details on the Monte Carlo simulation should be provided.*

Reply: We included a detailed description of the Monte Carlo simulations as Supplementary Note 8.

(6) *While it is quite useful to combine pump-probe results and time-resolved PL data to offer a unifying picture of excited state dynamics of (6,5) carbon nanotubes, how are the two data sets (Fig. 2) with very different timescales connected? Is the first 0.2 ns of the PL data used primarily for comparison with the pump-probe results, and how reliable is it (some description on the steps taken to remove the IRF would be useful).*

Reply: For all investigated nanotubes both types of transients were fitted by the same model function defined in eqn. 3. To account for the different instrument response functions (IRF) of the techniques, the fitted model functions were convoluted with the independently measured IRFs. In the case of the time-resolved PL measurement the IRF was determined by detecting the elastically scattered light from the sub-ps probe pulse at 1000 nm, the wavelength of PL emission. For the IRF of the TiSCAT experiment, we used the measured width of the sum-frequency generation signal between pump- and probe pulse on iron-iodate nanocrystals (Supplementary Note 5). The reviewer is right in pointing out the different timescales to which the two techniques are particularly sensitive. For all nanotubes we fitted first the PL transient to determine the slower exponential decay time as well as to obtain a first estimate of the faster diffusional time. We then used these parameters to fit the TiSCAT transient in the first few ps keeping the slower exponential decay time fixed. We then fixed the diffusional time and optimized the exponential decay using the PL transient and repeated this procedure until the parameters converged. We included the description of our approach in the Supplementary Note 5.

Reviewer #3 (Remarks to the Author):

The manuscript entitled "Probing the ultrafast dynamics of excitons in single semiconducting carbon nanotubes" by K. Birkmeier et al. describes the dynamics of excitons in a (6,5) single-walled carbon nanotube using time-resolved interferometric scattering spectroscopy and photoemission spectroscopy. So many studies on ultrafast dynamics of excitons in carbon nanotubes have already been reported. Nevertheless, the main advantage of this study is that the authors measured the dynamics

of excitons in a single carbon nanotube, which is a non-trivial work. Most of the previous works describe the dynamics of exciton of carbon nanotubes in in bulk state. Furthermore, they comprehensively investigated the exciton dynamics in single-wall carbon nanotubes, and they also adequately cited the previous studies for creating this manuscript. Therefore, the manuscript is of interest to nanotechnology or material science communities.

Here, I would like to raise several remarks, which may improve the quality of the manuscript.

1. They addressed that they measured 18 carbon nanotubes (page 8) and investigated 30 decay constants (Supplementary Figure 3). They also addressed that the exciton decay times vary from nanotube to nanotube because of defect density and nanotube length. However, they can estimate the nanotube length using their technique as shown in Fig. 1c. Therefore, at least, they can statistically understand the changes in exciton decay time affected by the nanotube length. Figure 3b with the information on nanotube length is essential for the further understanding of the dynamics of the excitons.

Reply: We thank the reviewer for this suggestion. We agree with the reviewer in that mono-exciton decay is expected to be faster for shorter nanotubes because of quenching at the ends. We determined the lengths of the nanotubes from the imaging data and plotted the derived values together with the corresponding exponential decay times in Supplementary Fig. 8. The obtained data points do not show a correlation between length and lifetime, however. This can be rationalized by the following arguments. All transients were detected at a position located in the center of the investigated nanotubes. With minimum nanotube lengths of 500 nm, which exceed the typical diffusion length reported for (6,5) SWCNTs on glass of about 100 nm (e.g. Science 316, 1465 (2007), Nano Lett. 10, 1595 (2010), Nano Lett. 12, 5091 (2012)), the influence of end quenching should be limited. In addition, the signal originates from a segment of the nanotube with a length of about 300 nm given by the spatial resolution of the experiment. This results in spatial averaging resulting further reducing the impact of local quenching. Moreover, even for the lowest excitation densities used in the experiment and needed to observe a clear signal in TiSCAT resulting in at least 2 initially generated excitons we found efficient exciton-exciton annihilation leading to rapid initial decay, which can be expected to dominate with respect to end-quenching. We added a brief discussion on the possible influence of the nanotube length to the revised version of the manuscript. **"In the present experiments TiSCAT and PL transients were recorded at the same position in the center of the investigated SWCNTs for direct comparability. No systematic influence of length**

dependent end quenching on the exponential decay time was observed because of a minimum nanotube length of 500 nm exceeding the exciton diffusion length of about 100 nm (Supplementary Fig. 8)."

2. *Related to 1, regarding the defect density, is it possible to derive the information about the defect density from the E11* or E11*- peaks from absorption or PL spectra? Is there any correlation?*

Reply: In case of the (6,5) nanotubes the PL peaks from E11* and E11*- are located at 1130 nm and 1250 nm, respectively (Nat. Commun. 12:2119 (2021)), outside of our detection window (Si-based CCD), unfortunately. The ensemble absorption spectra do not show significant defect contributions. Broadband ground state absorption spectra of single (6,5) on glass are extremely difficult to observe and would not allow for a quantitative analysis of defect-related states.

3. *They used two time-resolved spectroscopic measurements onto the (6,5) single-wall carbon nanotubes. They used time-resolved interferometric scattering spectroscopy in the early timescale (1 – 100 ps) and used time-resolved PL spectroscopy in the later timescale (0.1 – 1 ns). However, these two measurements do not temporally overlap together. For example, in Fig. 2b, the transients do not fully relax at 100 ps. Does this very slow decay correspond to the decay observed by time-resolved PL spectroscopy? If their setup allows them to measure extended time (~1 ns or ~30 cm), it is interesting to compare the signals of time-resolved interferometric scattering spectroscopy and photoemission spectroscopy.*

Reply: The two techniques cover the range of -2 to 100 ps (for TiSCAT) and -100 to 1.500 ps (time-resolved PL), respectively and thus overlap. Negative times are included to capture the signal rise upon pulsed excitation and to accurately consider the respective instrument response functions. While longer delay times are possible also in the case of TiSCAT the signal level achievable for single SWCNTs becomes too small for times exceeding 100 ps to quantitatively evaluate the transients. Because the observed exponential lifetimes are below ~50 ps with a majority below 20 ps, this does not represent a major limitation. In the case of PL transients, on the other hand, the temporal width of the instrument response function is 30 ps while the signal to noise ratio even for single nanotubes can readily reach 10^4 . This means that while ultrafast decay phenomena, such as EEA, cannot be accurately resolved by PL detection, the technique is ideally suited to explore excited state decay over

many orders of magnitude on the time-scale of 10 ps - ns. In our manuscript we exploit the complementary strengths of the two techniques using the procedure described Supplementary Note 5.

4. Is their discussion of exciton dynamics in a semiconducting nanotube valid for other carbon nanotubes with different chirality?

Reply: The presented experimental approach combining TiSCAT and time-resolved PL can be applied to other SWCNTs. To show this, we included recently obtained new data on single (6,4) SWCNTs in Supplementary Fig. 5. The obtained signal transients (Supplementary Fig. 6) can also be described by the model discussed in the main text. We also added the corresponding text to the main manuscript **“TiSCAT and PL images with similar contrast were also obtained for single (6,4) SWCNTs on glass (Supplementary Fig. 5) confirming the general applicability of the present experimental approach. In these experiments, the probe wavelength was tuned to 880 nm matching the E₁₁ resonance of (6,4) SWCNTs.”** and **“We further note that the fit function given by eqn. 3 can also be used to describe the transients recorded for (6,4) SWCNT on glass (Supplementary Fig. 6) indicating the general suitability of the model.”** The detection of (6,4) SWCNTs turned out to be more challenging compared to that of (6,5) nanotubes presumably due to a smaller absorption cross section. We thus utilized a tunable dual wavelength fiber laser with improved stability for excitation at 780 nm and probing at 880 nm, respectively. Unfortunately, this very stable laser system cannot be tuned to 1000 nm, the probe wavelength of (6,5) nanotubes.

5. In the abstract, they stated the segment length of the carbon nanotubes is 600 nm; however, in the main text, there is no information about the segment length.

Reply: We thank the reviewer for pointing this out. This segment length refers to the section of a carbon nanotube illuminated in the pump-probe experiment. This length is determined by the size of the diffraction-limited excitation spot corresponding to $\sim \lambda / \text{NA} = 880 \text{ nm} / 1.49 = 600 \text{ nm}$. We clarified this point in the revised version of the manuscript by adding the following text: **“The number of excitons specified on the upper x-axis in Fig.~3b corresponds to the total number of initially created excitons within the illuminated diffraction-limited nanotube segment length of $\sim \lambda / \text{NA} = 880 \text{ nm} / 1.49 = 600 \text{ nm}$.”**

6. Did the authors synthesize the (6,5) single-walled carbon nanotube? If yes, it is

better to show the synthesis procedure shortly or cite the article that shows the synthesis procedure. If they bought the sample, it is better to show the provider. They only showed how to wrap the carbon nanotubes with the polymer.

Reply: We thank the reviewer for the suggestion and added the corresponding text to the methods section of the manuscript. *"The SWCNT raw material, synthesized by the CoMoCAT procedure, was purchased from Sigma Aldrich (SG65)."*

7. (minor) They addressed the number of excitons as 2–15 in the abstract and 2–16 in summary. Please correct it.

Reply: We thank the reviewer and corrected the text accordingly.

REVIEWERS' COMMENTS

Reviewer #1 (Remarks to the Author):

Review comments for

Manuscript ID: NCOMMS-22-14484A

Title: Probing the Ultrafast Dynamics of Excitons in Single Semiconducting Carbon Nanotubes

Konrad Birkmeier, Tobias Hertel, and Achim Hartschuh

Submitted to: Nature Communications

Comments:

The importance of the work is still not clear. Perhaps this is not clearly written still. The current version pays too much focus on the methodology (TiSCAT compared to TA, TRPL) rather than new scientific findings. The new methodology would be important if the scientific outcomes are novel and important. Very less importance is given in the current version of the manuscript regarding new scientific findings, especially in the abstract, introduction, and conclusion.

The main novelty of this work is the investigations in a single SWCNT rather than CNT bundle, network, and suspension in the literature. In this case, different EEA and/or diffusion dynamics can be measured depending on the different types of SWCNTs, such as chiralities, diameter, edge-states, functional groups, etc. Such a distinction is missing in the literature for individual SWCNTs. If this can be demonstrated then, this work's novelty and importance would be significant.

However, the focus of this manuscript currently is on the TiSCAT majorly. Which is a marginal improvement compared to current literature unless new scientific findings are strongly demonstrated.

The only new scientific findings the authors mention is

“The new information in our manuscript is that exciton - exciton annihilation and the underlying diffusive transport can be observed in individual semiconducting SWCNTs in real-time and that the diffusional time can be predicted from the number of incident photons and the known diffusion coefficient.”

Then new figures (experimental data) of the real-time nature of the TiSCAT must be compared with TA and TRPL to clearly demonstrate the advantages and novelty of TiSCAT.

Authors further mentioned:

“Crucially, significant variations in the diffusional time from nanotube to nanotube are observed for the first time that indicate varying degrees of exciton localization.”

This is the most important scientific finding in the whole investigation. The importance of this should be clearly defined in the abstract, and introduction and should be demonstrated in the results, which is missing.

Moreover, as mentioned above, strong experimental demonstrations should be added to the manuscript from different types of SWCNTs, such as chiralities, diameter, edge-states, functional groups, etc. And the manuscript should be rewritten focusing on the results as well.

Then the manuscript should be important and acceptable for Nature Communication. Currently, this version of the manuscript is okay for other specific journals.

Reviewer #2 (Remarks to the Author):

The authors have adequately addressed the comments by the Reviewers and the manuscript is now suitable for publication.

Reviewer #3 (Remarks to the Author):

The manuscript entitled "Probing the Ultrafast Dynamics of Excitons in Single Semiconducting Carbon Nanotubes" by K. Birkmeier et al. describes excitonic dynamics in a single CNT using the combination of transient absorption and time-resolved photoluminescence spectroscopy. The authors answered all the questions raised in the previous communications. They also extended the experiments on the (6,4) CNTs. Therefore, I recommend that the manuscript be published in Nature Communications.

Review comments for

Manuscript ID: NCOMMS-22-14484A

Title: Probing the Ultrafast Dynamics of Excitons in Single Semiconducting Carbon Nanotubes

Konrad Birkmeier, Tobias Hertel, and Achim Hartschuh

Submitted to: Nature Communications

Comments:

The importance of the work is still not clear. Perhaps this is not clearly written still. The current version pays too much focus on the methodology (TiSCAT compared to TA, TRPL) rather than new scientific findings. The new methodology would be important if the scientific outcomes are novel and important. Very less importance is given in the current version of the manuscript regarding new scientific findings, especially in the abstract, introduction, and conclusion.

The main novelty of this work is the investigations in a single SWCNT rather than CNT bundle, network, and suspension in the literature. In this case, different EEA and/or diffusion dynamics can be measured depending on the different types of SWCNTs, such as chiralities, diameter, edge-states, functional groups, etc. Such a distinction is missing in the literature for individual SWCNTs. If this can be demonstrated then, this work's novelty and importance would be significant.

However, the focus of this manuscript currently is on the TiSCAT majorly. Which is a marginal improvement compared to current literature unless new scientific findings are strongly demonstrated.

The only new scientific findings the authors mention is

“ The new information in our manuscript is that exciton - exciton annihilation and the underlying diffusive transport can be observed in individual semiconducting SWCNTs in **real-time** and that the diffusional time can be predicted from the number of incident photons and the known diffusion coefficient.”

Then new figures (experimental data) of the **real-time nature** of the TiSCAT must be compared with TA and TRPL to clearly demonstrate the advantages and novelty of TiSCAT.

Authors further mentioned:

“Crucially, significant variations in the diffusional time from nanotube to nanotube are observed for the first time that indicate varying degrees of exciton localization.”

This is the most important scientific finding in the whole investigation. The importance of this should be clearly defined in the abstract, and introduction and should be demonstrated in the results, which is missing.

Moreover, as mentioned above, strong experimental demonstrations should be added to the manuscript from different types of SWCNTs, such as chiralities, diameter, edge-states, functional groups, etc. And the manuscript should be rewritten focusing on the results as well.

Then the manuscript should be important and acceptable for Nature Communication. Currently, this version of the manuscript is okay for other specific journals.

Reviewer #1 (Remarks to the Author):

The importance of the work is still not clear. Perhaps this is not clearly written still. The current version pays too much focus on the methodology (TiSCAT compared to TA, TRPL) rather than new scientific findings. The new methodology would be important if the scientific outcomes are novel and important. Very less importance is given in the current version of the manuscript regarding new scientific findings, especially in the abstract, introduction, and conclusion. The main novelty of this work is the investigations in a single SWCNT rather than CNT bundle, network, and suspension in the literature. In this case, different EEA and/or diffusion dynamics can be measured depending on the different types of SWCNTs, such as chiralities, diameter, edgestates, functional groups, etc. Such a distinction is missing in the literature for individual SWCNTs. If this can be demonstrated then, this work's novelty and importance would be significant.

Reply: We agree with reviewer in that the investigation of the chirality dependence of the EEA and/or diffusion dynamics as well as the influence of defects / dopants and functional groups would be of great interest. Such a comprehensive study, however, would clearly go beyond the scope of the present manuscript, which reports on the observation and modeling of the ultrafast exciton decay in single semiconducting nanotubes for the first time. In fact, the number of different nanotube chiralities is huge (virtually infinite), even after grouping them in families. We stress that we demonstrate in our manuscript that both the experimental technique and the model can be applied to a second nanotube chirality.

However, the focus of this manuscript currently is on the TiSCAT majorly. Which is a marginal improvement compared to current literature unless new scientific findings are strongly demonstrated. The only new scientific findings the authors mention is " The new information in our manuscript is that exciton - exciton annihilation and the underlying diffusive transport can be observed in individual semiconducting SWCNTs in real-time and that the diffusional time can be predicted from the number of incident photons and the known diffusion coefficient." Then new figures (experimental data) of the real-time nature of the TiSCAT must be compared with TA and TRPL to clearly demonstrate the advantages and novelty of TiSCAT.

Reply: This statement does not become clear to us. In our manuscript we note and show that the temporal resolution in time-correlated single photon counting used to observe the PL dynamics is significantly lower than that for the implemented

pump-probe scheme and with this is clearly not sufficient to resolve the early dynamics. The implemented pump-probe scheme is essentially a transient absorption measurement, as we discuss in our manuscript in detail and which becomes evident from the analysis of the signal formation. In the manuscript, we illustrate the experimental approach that enables the observation of single nanotubes focusing on the key aspects. The corresponding text covers less than two manuscript pages, compared to more than five pages for the discussion of the photo-physics of the nanotubes. Reducing this text further would significantly impair the readability of the manuscript, in our opinion. We do not agree with the reviewer in that the focus of the manuscript would be mainly on TiSCAT.

Authors further mentioned:

“Crucially, significant variations in the diffusional time from nanotube to nanotube are observed for the first time that indicate varying degrees of exciton localization.” This is the most important scientific finding in the whole investigation. The importance of this should be clearly defined in the abstract, and introduction and should be demonstrated in the results, which is missing.

Reply: We clearly demonstrate the variations of the diffusional times in Figure 3b. This finding is highlighted in the introduction and in the conclusion (page 5: “Comparison of data for different SWCNTs reveals a substantial variation of the diffusional time associated with EEA presumably due to different exciton 110 diffusion coefficients and / or exciton localization” and page 14 “Measurements on different nanotubes reveal a distribution of diffusional times”, respectively). In the abstract, we focus on the main challenges and results of the present study to match the length requirements.

Moreover, as mentioned above, strong experimental demonstrations should be added to the manuscript from different types of SWCNTs, such as chiralities, diameter, edge-states, functional groups, etc. And the manuscript should be rewritten focusing on the results as well. Then the manuscript should be important and acceptable for Nature Communication. Currently, this version of the manuscript is okay for other specific journals.

Reply: As noted above, we added the results on an additional nanotube chirality to the revised version. A comprehensive investigation of the chirality and diameter dependence as well as the role of different defect states and different functional groups would be well outside the scope of the present manuscript.